# Signal-Retaining Autophagy Indicator as a Quantitative Imaging Method for ER-Phagy

**DOI:** 10.3390/cells12081134

**Published:** 2023-04-11

**Authors:** Natalia Jimenez-Moreno, Carla Salomo-Coll, Laura C. Murphy, Simon Wilkinson

**Affiliations:** 1Cancer Research UK Scotland Centre, Institute of Genetics and Cancer, University of Edinburgh, Edinburgh EH4 2XR, UK; c.salomo-coll@sms.ed.ac.uk (C.S.-C.); s.wilkinson@ed.ac.uk (S.W.); 2MRC Human Genetics Unit, Institute of Genetics and Cancer, University of Edinburgh, Edinburgh EH4 2XU, UK; laura.murphy@ed.ac.uk

**Keywords:** autophagy, fluorescent protein, reporter, SRAI, autophagy flux, lysosome, ER, ER-phagy

## Abstract

Autophagy is an intracellular lysosomal degradation pathway by which cytoplasmic cargoes are removed to maintain cellular homeostasis. Monitoring autophagy flux is crucial to understand the autophagy process and its biological significance. However, assays to measure autophagy flux are either complex, low throughput or not sensitive enough for reliable quantitative results. Recently, ER-phagy has emerged as a physiologically relevant pathway to maintain ER homeostasis but the process is poorly understood, highlighting the need for tools to monitor ER-phagy flux. In this study, we validate the use of the signal-retaining autophagy indicator (SRAI), a fixable fluorescent probe recently generated and described to detect mitophagy, as a versatile, sensitive and convenient probe for monitoring ER-phagy. This includes the study of either general selective degradation of the endoplasmic reticulum (ER-phagy) or individual forms of ER-phagy involving specific cargo receptors (e.g., FAM134B, FAM134C, TEX264 and CCPG1). Crucially, we present a detailed protocol for the quantification of autophagic flux using automated microscopy and high throughput analysis. Overall, this probe provides a reliable and convenient tool for the measurement of ER-phagy.

## 1. Introduction

Autophagy is a catabolic process by which cytosolic cargoes are degraded by lysosomes to maintain cellular homeostasis. The most well-studied form of autophagy, also known as macroautophagy, requires the formation of an autophagosome that will eventually fuse with lysosomes, forming an autolysosome, wherein the contents will be degraded and macromolecules will be recycled in the cell [1,2,3]. Autophagy can be non-selective, also known as bulk autophagy, or selective, wherein specific organellar cargoes are recognised by “cargo receptors” that bind to the nascent autophagosome. Numerous selective autophagy pathways have been described, generally named after the cargo that is degraded—e.g., mitophagy for mitochondrial degradation and ER-phagy for endoplasmic reticulum (ER) degradation (for a review of the most recently identified selective autophagy pathways see [4]). Thus, autophagy is crucial for cellular function and indeed dysregulations in this pathway are associated with diverse pathologies such as neurodegenerative diseases and cancer [2,5,6,7,8]. 

Given the important role of autophagy in health and disease, mammalian autophagy research has been the focus of numerous studies. Work in this field has increased exponentially during the last decade [9]. A major focus is to find methods to measure activity of the pathway (for a most recent review of guidelines to monitor autophagy see [10]). In particular, the measurement of autophagic flux is desirable (i.e., measuring completion of the pathway via delivery of cargo to the lysosome) [11]. Typically, flux is measured by monitoring the turnover rate of the autophagosomal protein microtubule-associated protein 1 light chain 3 (LC3) or autophagy adaptors such as sequestosome-1 (p62) [11]. This can be achieved using tandem fluorescent-based reporters, in most cases mRFP (red fluorescent protein) coupled with GFP (green fluorescent protein). In this system, GFP is quenched in autolysosomes due to the low pH in this compartment [12,13,14,15]. Thus, for the best characterized autophagy reporter, mRFP-GFP-LC3 [15], the reporter fluoresces both green and red in the cytosol or when recruited to autophagosomes. However, after fusion of autophagosomes with the lysosome (i.e., autolysosome) only the RFP is detectable. This way, autophagic flux can be observed by the loss of the GFP fluorescence relative to RFP. This is often quantified by counting individual punctate signals (i.e., autophagosomes and autolysosomes) in a relatively low throughput process involving manually driven fluorescence microscopy. However, the nature of these two fluorophores (intramolecular resonance transfer and incomplete resistance to acidity) reduces the assay sensitivity and affects quantification [16,17]. 

To address this problem, Katayama et al. described a sensitive and quantifiable mitophagy reporter based on the acid-stable fluorescent protein Keima, also known as mt-Keima, which emits different colored signals depending on the environmental pH. However, it can only be used for live imaging as it exhibits reversible acid sensitivity [18]. To overcome this, they recently developed a new “signal-retaining autophagy indicator” (SRAI). This is a pH sensitive and quantifiable probe that can be read out in both living and aldehyde-fixed specimens, which allows higher throughput screening assays [17]. SRAI is composed of an acid-insensitive CFP (cyan fluorescent protein) mutant, named TOLLES (TOLerance of Lysosomal EnvironmentS), optimized to retain full fluorescence at low pH, and a highly acid-sensitive YFP (yellow fluorescent protein) variant named YPet, that is irreversibly acid-denatured [17,19]. In this study, they fused SRAI to the COX VIII presequence, driving mitochondrial localization [17], thus developing the mito-SRAI reporter for mitophagy flux measurement. In addition, they defined a parameter called the mitophagy index, which was calculated as the quotient of the area of lysosomal signal relative to total cell area in fluorescence cell images. They established this to be a reliable metric for mitophagic flux [17]. However, the general applicability of the SRAI probe to study bulk autophagy or other types of selective autophagy was not assessed. 

In this paper, we present a basic characterization of SRAI as a sensitive and quantifiable probe for the study of ER-phagy flux, and use this to highlight several important considerations when studying this process. In particular, we present the distinction between measurement of general ER-phagy flux and the turnover of individual ER-phagy receptors. In addition, we describe a detailed quantification protocol for the measurement of the TOLLES:YPet index (in analogy to the aforementioned mitophagy index). This reads out overall ER-phagy flux, or flux of individual ER-phagy receptors to the lysosome. This quantification method deals with the particular problems of sensitivity that may be encountered using a probe that is distributed abundantly throughout the cytosolic area, on or within the sheet and/or tubular ER networks, only a modest proportion of which may be transported to the lysosome during the experimental period. Overall, we conclude that SRAI-based reporters present a reliable, easy and high throughput method for measurement of ER-phagy activity.

## 2. Materials and Methods

### 2.1. Cell Culture and Cell Lines

Pancreatic ductal adenocarcinoma (PDAC) KPC mouse cell line (C57/BL6 genetic background, obtained from Dr. Owen Sansom, Beatson Institute for Cancer Research, Glasgow, UK), and HEK293FT human embryonic kidney cell line (Clontech, Mountain View, CA, USA), were maintained in high-glucose DMEM medium (Thermo Scientific, Scotland, UK, 10313021) supplemented with 10% FBS (Thermo Scientific, 10270-106), 2 mM L-glutamine (Thermo Scientific, 25030081) and 10 U/mL penicillin/streptomycin (Thermo Scientific, 15140122) at 37 °C in 5% CO_2_. Amino acid starvations were performed by washing the cells three times in phosphate-buffered saline (PBS) before culture in fresh Earle’s balanced salt solution (EBSS, Sigma, Scotland, UK, E2888) for 4 h. For lysosomal block, cells were treated with Bafilomycin A1 (BafA1, Cayman, Ann Arbor, MI, USA, 11038–500) at 50 nM for 4 h in full DMEM or EBSS media. 

### 2.2. Plasmids and Molecular Cloning

To overexpress the CRISPR associated protein 9 (Cas9), we used the commercially available SpCas9 hygro lentiviral vector (Addgene, MA, USA, 104995, a gift from Brett Stringer [20]). pCR-Blunt II-TOPO with U6 polymerase promoter was used for sgRNAs expression (a gift from Dr. George Church, Harvard University, Cambridge, MA, USA, [21], also available from Addgene, 41824). Molecular cloning of the sgRNAs was performed as described in the synthesis protocol (Addgene, 41824, Option B). 

To overexpress the SRAI sequence (i.e., TOLLES and YPet linked by triple repeat of the amino acid linker Gly-Gly-Gly-Gly-Ser [22]), cDNA sequence based on [17] was synthetized codon optimized for *M. musculus* (IDT) and cloned into pLX304 DEST vector (Addgene, 25890, a gift from David Root [23]) to generate a pLX304 SRAI N1 DEST and pLX304 SRAI C1 DEST lentiviral vectors. From this plasmid, molecular cloning was used to introduce an N-terminal ER signal sequence (ss) and the C terminal retaining ER sequence KDEL [24] to create the pLX304 ss-SRAI-KDEL. In addition, for the fusion of the autophagy and ER-phagy cargo receptors to the tandem tag fluorescent probe we performed LR reactions according to manufacturer’s instructions (Thermo Scientific, 11791020). Plasmids used for these reaction were: pDONR223 *LC3B* (from [25]) to generate SRAI-LC3B; pDONR223 *Fam134b* (Source Bioscience, Nottingham, UK, NM_001034851.2) to generate SRAI-FAM134B; pDONR223 *Fam134c* (Source Bioscience, NM_026501.3) to generate SRAI-FAM134C; pDONR223 *Ccpg1* (Source Bioscience, NM_001114328.2) to generate SRAI-CCPG1 and pDONR223 *Tex264* (Source Bioscience, BC002248.2) to generate TEX264-SRAI. Detailed sequence maps are available from authors upon request.

Alternatively, to monitor ER-phagy flux, the lentiviral vector pCW57-CMV ss-mRFP-GFP-KDEL (Addgene, 128257, a gift from Noburo Mizushima [24]) was used. 

### 2.3. Viruses, Transduction and Stable Cell Lines

Lentivirus were produced in HEK293FT cells by transient transfection using lipofectamine 2000 (Thermo Scientific, 11668019), 1 µg of the plasmid of interest was transfected together with 0.75 µg of the packing plasmid pAX2 and 0.25 µg of the envelope plasmid pMGD2 (Addgene, 12259 and 12260, gifts from Didier Trono). Viruses were harvested in high-serum DMEM (30% FBS) 48 h after transfection. Media was collected and filtered with a 0.45 µm filter.

PDAC cells overexpressing the different constructs were generated as follows: cells were plated in 6 cm dishes and transduced with the corresponding lentivirus in the presence of 10 µg/mL polybrene (Sigma, TR-1003). After two days, cells were fed with media supplemented with either 1 mg/mL hygromycin B (Sigma, 400052) for 10 d—for cells expressing Cas9:hygro—or with 10 µg/mL blasticidin (Thermo Scientific, 155528) for 4 d—for cells expressing the SRAI constructs—or with 3 µg/mL puromycin (Sigma, P6255) for 2 d—for cells expressing ss-mRFP-GFP-KDEL. These concentrations were selected using a titration kill curve in PDAC cells. 

Prior to the experiment, PDAC cells stably expressing ss-mRFP-GFP-KDEL were treated with 0.5 µg/mL doxycycline for 24 h to induce the expression of the construct. 

### 2.4. Generation of ΔAtg7 PDAC Cell Line Using CRISPR/Cas9 Gene Editing

Δ*Atg7* PDAC cell line was generated using CRISPR/Cas9 technology [21]. sgRNA targeting *Atg7* (5′-GAAACTTGTTGAGGAGCAT-3′, [26]) or luciferase (5′-ACGGCGGCGGGAAGTTCAC-3′, [27], referred to as sgControl) were co-transfected in a ratio 3:1 with a plasmid expressing puromycin resistance (Addgene, 62988, a gift from Feng Zhang [28]) using lipofectamine 2000. The following day, cells were fed with media supplemented with 3 µg/mL puromycin (Sigma, P6255) for 2 d. Cells were passaged at least once before performing experiments and no more than 5 times. 

### 2.5. Generation of SRAI-Expressing PDAC Cell Lines Using FACS Sorting

PDAC Cas9 cells stably expressing SRAI reporters were created by infection of PDAC with pLX304 SRAI lentivirus and selection with blasticidin (10 µg/mL, 4 d). Then, 10 × 10^6^ cells/mL were harvested in FACS sorting buffer (non-phenol red DMEM [Thermo Scientific, 31053028] supplemented with 1% FBS, 2 mM L-glutamine and 10 U/mL penicillin/streptomycin). Cells were sorted using a FACSAria II SORP (BD Bioscience, New Jersey, NJ, USA). TOLLES was excited by a 405 nm laser line and its emission was collected through 525/50BP; YPet was excited by a 488 nm laser line and its emission was collected through 525/50BP. First, single cells were gated using FSC-Height (FSC-H) by FSC-Area (FSC-A). Then, cells were gated for a narrow range of YPet and TOLLES expression and cells were collected, and data were recorded using BD FACSDiva 6.3.1 (see Appendix A as an example). Cells were then cultured in full DMEM and incubated at 37 °C in 5% CO_2_.

### 2.6. Cell Imaging by Wide-Field Microscopy and Automated Acquisition

PDAC Cas9 cells stably expressing SRAI constructs were plated on glass bottom 96 well plates (IBL, Gerasdorf, Austria, P96-1.5H-N) to reach 70–90% confluency (approximately 20,000 cells). The following day, cells were then starved (EBSS) or treated with 50 nM BafA1 for 4 h in full nutrient or starvation conditions. After the corresponding times, cells were fixed by incubating with 4% paraformaldehyde in PBS, pH 7.2 at room temperature for 10 min. Cells were then washed twice with PBS and kept at 4 °C overnight. Cells immersed in PBS were imaged 1–3 days after fixation.

Images were captured with a Nikon Eclipse Ti-E inverted microscope (Nikon Instruments UK Ltd., Kingston Upon Thames, UK) using a 40X NA 0.75 objective. Widefield illumination is provided by a CoolLED PE-4000 LED light source (CoolLED Ltd., Andover, UK) combined with LED-CFP/YFP/mCherry-3X-A (Pinkel) filter sets (IDEX Health & Science, LLC, Center of Excellence, Rochester, NY, USA). Images were acquired with a Photometrics Prime BSI sCMOS camera (Teledyne Photometrics, 3440 E. Britannia Drive, Tucson, AZ, USA). Image bit depth was kept at 12-bit (CMS gain) and the exposure time was around 100–500 ms depending on the fluorescence levels. 

Conditional imaging was setup and controlled using the Nikon Nis-Elements JOBS module (see Appendix A as an example). The JOB (script available from GitHub [29]) incorporates automated two wavelength capture of 3D images (Z stacks of 1.5 µm per slide for CFP and YFP filters) from an array of imaging sites (XY positions) using a motorised XY stage within user selected wells (a total of 6–8 fields per well with around 60–100 cells per field were imaged) of a 96 well plate. Images are shown as z-projects of 5 z-steps. 

For PDAC cells stably expressing ss-RFP-GFP-KDEL, cells were cultured in the presence of doxycycline (0.5 µg/mL, 24 h). Then, doxycycline was removed and cells were treated with the indicated treatments. Images were captured using the GFP and TRITC filters. 

### 2.7. Imaging Quantification of the TOLLES:YPet Index

TOLLES:YPet index was obtained using a developed ImageJ [30] macro based on the mito-SRAI ratio reported in [17] (Appendix A, also available from GitHub [26]). A step-by-step protocol is described in the Results section (see Appendix A). Briefly, this macro uses the images obtained from the wide-field microscope and calculates the TOLLES:YPet index using different thresholds (chosen by the user) (Appendix A). Furthermore, to then create independent files with the values using different thresholds we generated a post-analysis Python script (Appendix A). 

### 2.8. Immunocytochemistry and Confocal Imaging

Immunofluorescence was performed as previously described [31]. Briefly, parental PDAC cells or PDAC ss-SRAI-KDEL stable cells were seeded on 12 mm coverslips (Paul Marienfeld GmbH, Stuttgart, Germany, 0117520). The following day cells were treated with the corresponding treatments and then incubated with 4% formaldehyde for 10 min. Formaldehyde-fixed cells were incubated with 0.25% Triton X-100 in PBS for 15 min and then incubated with blocking solution containing 5% *w*/*v* bovine serum albumin (BSA) for at least half an hour at room temperature. Cells were then incubated 2 h with α-LAMP1 (Abcam, Cambridge, UK ab25245) or α-Calnexin (Proteintech, Manchester, UK, 101427-2-AP) or α-FAM134B (CST, Leiden, The Netherlands, 83414) primary antibodies prepared in 2.5% BSA. Cells were washed three times with PBS and incubated with the secondary antibodies (IgG H+L AlexaFluor 488 or IgG H+L AlexaFluor 647, A-31573 for SRAI-expressing cells) for 1 h. Cells were then washed again with PBS, mounted in slides in Dako fluorescent mounting medium (Agilent, Stockport, UK, S3023) and fluorescence images were acquired using a 60× oil lens on the multimodal Imaging Platform Dragonfly (Andor technologies, Belfast, UK), with 405, 488 and 640 nm lasers built on a Nikon Eclipse Ti-E inverted microscope body with Perfect focus system (Nikon Instruments, Tokyo, Japan). 

### 2.9. Immunobloting

Immunoblotting was performed as previously described [31]. Briefly, PDAC cells were lysed in SDS lysis buffer (4% SDS, 150 mM NaCl, 50 mM Tris, pH 7.5). All samples were diluted with Laemmli buffer to 1 x before heating at 95 °C for 5 min before gel electrophoresis. Samples were separated using SDS-PAGE employing MOPS NuPAGE 4–12% gels from Invitrogen as directed by the manufacturer. Proteins were transferred to nitrocellulose membranes. Membranes were blocked with 5% BSA in TBST and then incubated with α-ATG7 (CST, 8558) or α-α-tubulin (CST, 3873) primary antibodies diluted in 2.5% BSA in TBST + 0.05% sodium azide overnight. For secondary antibodies, membranes were incubated with secondary HRP-linked antibodies (CST) diluted 1:4000 in 2.5% BSA in TBST. Blots were developed using X-ray film with standard ECL.

### 2.10. Image Analysis and Statistical Analysis

TOLLES:YPet index was calculated as described above. The number of reticulolysosomes formed in ss-mRFP-GFP-KDEL cells were determined by counting GFP quenched, mRFP positive foci using the ImageJ MitoQC plug-in [14]. Graphical results were analyzed with GraphPad Prism 9 (GraphPad Software, San Diego, CA, USA), using a one-way ANOVA followed by Holm-Šídák post hoc test. Results are represented as mean ± SD, *n* = 3.

## 3. Results

### 3.1. Characterization of SRAI as a Probe to Observe Bulk Autophagic Flux, Extending the Utility of This Probe beyond Mitophagy

Katayama et al. [17] developed the mito-SRAI reporter, based on the fluorescent proteins TOLLES and YPet, in order to measure mitophagy flux. However, a general reporter for bulk autophagic flux was not generated, so the generalizability of this technique was unclear. In order to address this, we tested the validity of this probe to monitor bulk autophagy. We fused SRAI to the autophagosomal membrane protein LC3B (Figure 1A,B). Apart from the marked difference in acid-sensitivity between the two fluorophores, the CFP-variant TOLLES acts as a FRET donor and the YFP-variant YPet as a FRET acceptor [17]. Thus, in a neutral environment (e.g., phagophore and autophagosomes), both fluorophores are present, but the apparent TOLLES emission is weak. However, in autolysosomes, YPet is quenched thus, relieving FRET and producing a big shift towards TOLLES fluorescence [17] (Figure 1A,B). In addition, because the signal is not sensitive to fixation, we used paraformaldehyde-fixed, multiwell-arrayed samples, employing conditional epifluorescence imaging with a motorised stage to obtain multiple z-stack images per sample (Appendix A). Images were taken of murine pancreatic ductal adenocarcinoma (PDAC) cells, with several fields per condition being captured (Appendix A). However, heterogeneity in expression was apparent in these images. It was expected that this would confound automated downstream analysis to extract a reliable parameter of autophagic flux. To avoid the need to generate single cell clones, which is prone to introducing artefacts into cellular systems, we instead created a pool of stable SRAI-LC3B-expressing cells having a relatively narrow range of expression of the construct (based upon YPet and TOLLES fluorescence, Appendix A). The presence of autophagosomes (TOLLES and YPet-positive puncta) and autolysosomes (TOLLES-positive and YPet-negative puncta) was observed in full nutrient conditions, indicative of the basal LC3 turnover rate in these cells (Appendix A). Importantly, a dramatic increase in autolysosomes was observed in nutrient starvation conditions (EBSS), a known inducer of bulk autophagy (Figure 1C). This was ablated in the presence of the lysosomal inhibitor Bafilomycin A1 (BafA1) or by deletion of *Atg7*, a core autophagy gene required for LC3B attachment to autophagosome membranes (Δ*Atg7* PDAC cells). This was seen as an increase in autophagosomes at the expense of autolysosomes, or a complete absence of SRAI-LC3B foci, respectively (Figure 1C). These results indicate that the previously reported SRAI-based assay can be adapted to monitor bulk autophagic flux. These results also mirror those that would be expected for the current gold-standard, aldehyde-fixable mRFP-GFP-LC3 reporter [15,32].

### 3.2. Establishing a Methodology for Semi-Automated Quantification of Autophagic Flux

In order to quantify autophagic flux using the SRAI-LC3B reporter, we developed an ImageJ macro (Appendix A). This calculates on a per-field basis the quotient of the area of pixels with a TOLLES:YPet ratio above an arbitrary threshold over the total area that the SRAI fluorophore occupies within the field. We name this metric the TOLLES:YPet index. It is analogous to the previously reported mitophagy index used to measure mitophagy with mito-SRAI [17]. The higher the index, the higher the autophagy flux is interpreted as being. The macro uses z-stack images as input (we find that z-stack images are best used for automated imaging by widefield epifluorescence microscopy to ensure that the best plane is being captured). It then calculates multiple TOLLES:YPet indices using different thresholds of TOLLES:YPet ratio to identify pixels that contribute to the numerator (usually thresholds 0.5–4). User input is needed to: decide the z-slice to be analyzed (i.e., the image where the cells are in the best focal plane); reject any images from the quantification (e.g., poor image quality or no signal); select the best background region of interest to threshold the images (Appendix A). As an output, the macro will generate ratio-mapped images representing the TOLLES/YPet ratio levels across the field along with the TOLLES:YPet indices at the different thresholds for these ratios (Appendix A and Figure 2A). 

Importantly, we observed higher ratios across the ratio-mapped images when there was relatively more TOLLES fluorescence in the cells (i.e., EBSS-starved cells) compared to autophagy block via BafA1 or *Atg7* deletion (Figure 2A). Correlating with these images, TOLLES:YPet index quantifications using 3 different thresholds showed a significant increase in autophagic flux in starved cells (EBSS), and by contrast, BafA1 treatment or *Atg7* deletion reduced the TOLLES:YPet indices (Figure 2B). For final data presentation, one has to select one particular threshold across the experiment, to yield one set of TOLLES:YPet indices (one per condition, Appendix A). The selected threshold will be experiment- and reporter-specific (i.e., SRAI-fused to different targeting moieties). Thus, we recommend using positive and negative controls such as EBSS-starvation and BafA1, for example, to empirically determine the threshold that provides optimal separation of signals and lowest variances.

Finally, the original pool of non-FACS-sorted SRAI-LC3B cells were also tested for quantification using this methodology. Although the same trend was observed under starvation conditions and autophagy block, the separation of control conditions and variances were significantly improved by using FACS-sorted cells (Appendix A), validating our initial qualitative judgement on the need for relatively homogenous expression across the population of reporter cells.

### 3.3. Development of ss-SRAI-KDEL for Reliable Measurement of General ER-Phagy

The endoplasmic reticulum (ER) is a ubiquitous and dynamic organelle, subject to turnover and remodeling to ensure optimal function. Crucially, the ER is removed by a selective form of autophagy, known as ER-phagy, or reticulophagy. This is a fundamental pathway in the maintenance of ER homeostasis/cellular health. ER-phagy is induced in response to different stimuli such as starvation or the activation of the unfolded protein response (UPR) [31,33,34]. Overall ER-phagy flux is usually monitored using ER luminal or ER membrane proteins fused to the tandem mRFP-GFP probe [35]. However, this presents the same sensitivity limitations for reliable quantitative results as discussed in the Introduction. To confirm this, we used the ER luminal reporter ss-RFP-GFP-KDEL (Figure 3A). Here, mRFP-GFP is fused with an N-terminal ER signal sequence (ss) from binding immunoglobulin protein (BiP) and a C-terminal ER retention sequence KDEL [36]. We observed an increase in the number of reticulolysosomes (RFP-positive and GFP-negative puncta) under starvation conditions and this was blocked with BafA1, in line with published data (Figure 3A) [36]. 

To confirm the potential of the SRAI probe to study overall ER-phagy, we similarly targeted the SRAI probe to the ER lumen (i.e., fusion of the N-terminal ER signal sequence (ss) and the C-terminal ER retention sequence KDEL) (Figure 3B,C). Importantly, we confirmed the ER localization of this new reporter, referred to as ss-SRAI-KDEL, by co-staining with the ER marker Calnexin (Figure 3D). Crucially, we also observed colocalization of TOLLES-positive and YPet-negative puncta with the lysosomal marker lysosomal-associated membrane protein 1 (LAMP1) under starved conditions (Figure 3E). In addition, ER-phagy flux was significantly increased in starved cells (EBSS) in a manner dependent on autophagy; this increase was lost in Δ*Atg7* cells or upon BafA1 treatment (Figure 3F,G). By contrast, BafA1 on its own did not have an effect on basal ER-phagy flux. This could be explained by the low flux levels in the full nutrient condition as shown by very few TOLLES-positive and YPet-negative puncta in the cells, correlating with a small portion of the ER being targeted to the lysosome relative to the whole ER network under basal conditions (Figure 3F,G). Indeed, these results are in line with those using ss-mRFP-GFP-KDEL (Figure 3A). However, we believe that the ss-SRAI-KDEL SRAI reporter has an advantage as it is sensitive enough to semi-automatedly derive a TOLLES:YPet index from rapid analysis of numerous fields of cells [17]. This allows acquisition of statistically robust data, as well as not being affected by relatively high background levels in the channel for the pH-resistant signal (as seen for the red channel when imaging mRFP-GFP, Figure 3A). Thus, here we present the ss-SRAI-KDEL reporter as a novel indicator for the analysis of general ER-phagy, quantitatively and reliably.

### 3.4. ER-Phagy Flux Associated with Specific ER-Phagy Receptors Can Be Detected Using the SRAI Probe

Selective autophagy relies on the presence of autophagy cargo receptors that will recognize both the cargo and the autophagosomal membrane [37]. Particularly, for ER-phagy, several ER-phagy receptors have been characterized [38]. ER-phagy receptors are generally ER membrane-bound proteins (either reticulon homology domain (RHD)-or transmembrane domain (TMD)-containing proteins) or cytosolic proteins that will recognize an ER membrane protein [39]. They also bind to nascent autophagosomes via LIR motifs (LC3 interacting region motifs), and in the case of cell cycle progression 1 (CCPG1), can also recognize the autophagy-initiation protein FAK family kinase-interacting protein of 200 kDa (FIP200) (Figure 4A). Importantly, it is emerging that while general ER-phagy flux captures some aspects of ER-phagy, it is highly likely that localized flux of individual receptors plays different roles in remodeling the overall ER, either in controlling morphology or via removal of specific ER content. Furthermore, different upstream signalling pathways trigger the activation of different receptors [40]. This highlights the need to not just measure general autophagy/ER-phagy flux but also to compare the sensitivity of the different receptors to different stimuli.

Here, we selected some examples of ER-phagy cargo adaptors in order to study ER-phagy receptor flux by fusion with the SRAI probe. These were: family with sequence similarity 134B (FAM134B, also known as RETREG1); FAM134C (also known as RETREG3); testis expressed gene 264 (TEX264); CCPG1 (Figure 4A,B) [24,31,41,42]. First, we observed that FAM134B flux in stable PDAC cells expressing SRAI-FAM134B was significantly increased in EBSS-starvation conditions, and this was reverted after lysosomal block (BafA1) or when *Atg7* was deleted (Figure 4C). In addition, FAM134B-degradation levels in basal conditions were relatively high as shown by the presence of TOLLES-only puncta (TOLLES-positive and YPet-negative), consistent with published data indicating a role of FAM134B in basal conditions [41]. Crucially, to confirm that endogenous FAM134B flux follows a similar trend to the overexpressed reporter, we looked at FAM134B localization in PDAC cells upon BafA1 treatment (Figure 4D). Indeed, FAM134B puncta accumulation after autophagy block indicates high basal turnover rate of the receptor in line with our observation with the SRAI-FAM134B reporter.

Similarly, analysis of the degradation of its paralog FAM134C using the SRAI-FAM134C reporter showed an increase in the TOLLES:YPet index under nutrient deprivation and this ratio was significantly reduced after BafA1 and in Δ*Atg7* cells (Figure 5A). Importantly, unlike FAM134B where BafA1 treatment considerably reduced the basal flux, the basal FAM134C-associated TOLLES:YPet index was not affected by autophagy block. This highlights the low FAM134C-degradation levels in basal conditions (i.e., low TOLLES-positive and YPet-negative puncta), consistent with previously reported studies [41,42,43,44]. 

To extend the use of this probe to monitor the degradation of other ER-phagy receptors, we analyzed the degradation of single type I and type II transmembrane ER proteins TEX264 and CCPG1 using TEX264-SRAI and SRAI-CCPG1 stable cell lines. For both, an increase in TOLLES:YPet indeces was observed under starved conditions and this was dependent on autophagy, as shown by the reduction in the indices after BafA1 or in autophagy deficient cells (Figure 5B,C), in line with recently published studies [24,31,36]. 

In conclusion, ER-phagy can be studied using the SRAI probe highlighting the potential of SRAI to study other types of selective autophagy apart from mitophagy [17]. Particularly for the study of selective autophagy where only a minority portion of the probe might be degraded, other probes like the tandem mRFP-GFP reporter are more limited. This is due to the use of low throughput quantification methods (e.g., mitoQC counter [14]) and/or sensitivity issues as discussed in the Introduction. In this scenario, the TOLLES:YPet index also has the potential to extend the use of these reporters for high-content screening and the capacity to look at small changes associated with physiologic stimuli or inhibiting basal fluxes. Crucially, this might be used for the characterization of novel regulators of the ER-phagy pathway or for drug screening and therapeutic development. 

## 4. Discussion

The study of autophagic flux in mammalian cells is a prominent focus in autophagy research [11,12]. However, autophagic flux assays can be time consuming or difficult to quantify. To overcome this, major advances have recently been reported, including mito-SRAI [17] and most recently, the use of pulse-chasable Halo-based processing assays for quantitatively monitoring autophagy [45,46]. In this study, we focus on the use of the SRAI probe as a sensitive and convenient fluorescent probe suitable for high-content assays, for the study of general ER-phagy or specific forms of ER-phagy. To this date, the study of ER-phagy flux has been focused on the use of mRFP-GFP reporters or most recently the Halotag-based reporters suitable for lower throughput analysis when looking at small changes in flux [35,45]. Here, we show that the SRAI probe is an effective way to quantitatively monitor ER-phagy under different conditions by analyzing the TOLLES:YPet index. In addition, although we have not addressed this in this study, this probe might be used to quantify the number of lysosomes versus the number of autophagosomes to provide additional information as on autophagosome formation versus lysosomal block. In addition, although not yet tested, this probe also has the potential to be used in vivo like similar reporters employing the mRFP-GFP probe (e.g., ss-mRFP-GFP-KDEL and mRFP-GFP-FAM134C [43]).

On the other hand, this assay has some limitations, in common with many widely-used autophagic flux assays, such as: the need for overexpression of the exogenous reporter, although potentially, CRISPR knock in technology could be used to overcome this [47]; the potential accumulation of TOLLES in lysosomes over time and the possibility of affecting lysosomal activity; the use of a bulky tag that can potentially affect protein function or induce aggregation [48]. Recently described Halo-tag probe-based assays for autophagic flux may circumvent the latter consideration to some degree, but the Halo-tag is still relatively bulky compared to small epitope tags, and these probes are relatively low throughput [45].

Furthermore, analysis of selective autophagy of the ER can be difficult as only a small but significant fraction of the ER may be degraded under certain stimuli unlike bulk autophagy (i.e., as measured by LC3B-based reporters where relatively high levels of autophagy flux are observed). However, with this reporter, we show that a reliable measurement is possible, validating the potential of this probe to measure flux in a more challenging scenario and highlighting the sensitivity of this reporter. 

In addition, recent data suggests that ER-phagy cargo receptors respond to different upstream signals. They cooperate with quality-control pathways such as ER-associated protein degradation (ERAD) or UPR [35,40], and these SRAI-based reporters could be used to explore these stimuli. In this study, we used different ER-phagy receptors to analyze ER-phagy receptor flux and we observed that they all responded to nutrient starvation as previously reported [24,31,36,41,42,43,44]. Furthermore, we also observed some differences between the receptors such as the low basal degradation levels of the ER-phagy receptor FAM134C in comparison to FAM134B as previously reported [43]. We anticipate that studying individual forms of ER-phagy involving specific cargo receptors will help determine the different role of the receptors in ER-phagy and help develop our limited knowledge of ER-phagy. 

Overall, this probe provides a reliable tool for the measurement of ER-phagy flux with potential for higher throughput and can be used to help understand this pathway and its biological significance.

## Figures and Tables

**Figure 1 cells-12-01134-f001:**
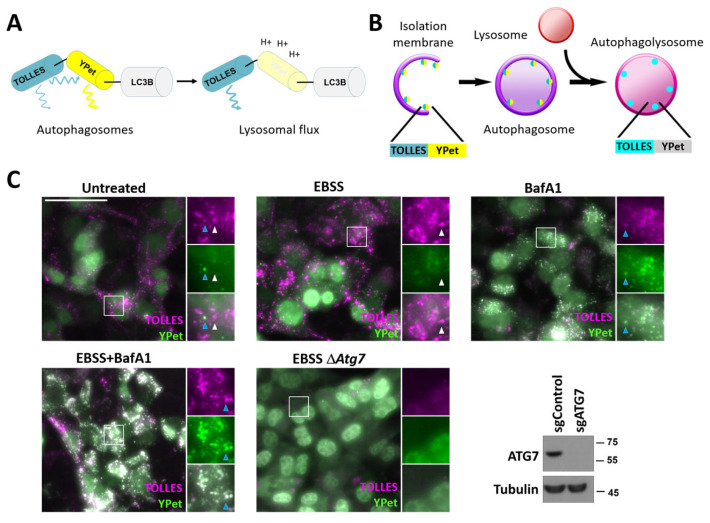
Characterization of SRAI as a reporter to observe bulk autophagic flux. (**A**) Schematic of the expected fluorescence profile of SRAI (TOLLES-YPet)-LC3B within different compartments (representing TOLLES as a FRET donor and YPet as a FRET acceptor). TOLLES is represented in blue and YPet in yellow or faint yellow if quenched. (**B**) SRAI-LC3B reporter localizes in phagophores and autophagosomes and TOLLES and YPet fluorescence is observed. However, after lysosomal fusion, the YPet is quenched and only TOLLES fluorescence is visible. (**C**) Imaging-based autophagic flux assay in SRAI-LC3B PDAC cells (transfected with sgControl or sg*Atg7*) in full nutrients (untreated) or following 4 h starvation (EBSS) in the presence or absence of BafA1 (50 nM, 4 h). Representative immunofluorescence images of TOLLES (magenta) and YPet (green); white arrows show examples of TOLLES-positive and YPet-negative puncta and blue arrows indicate TOLLES and YPet-positive puncta. Bar = 50 µm. On the bottom right, immunoblot showing ATG7 levels in CRISPR knockout Δ*Atg7* PDAC cells in comparison to control.

**Figure 2 cells-12-01134-f002:**
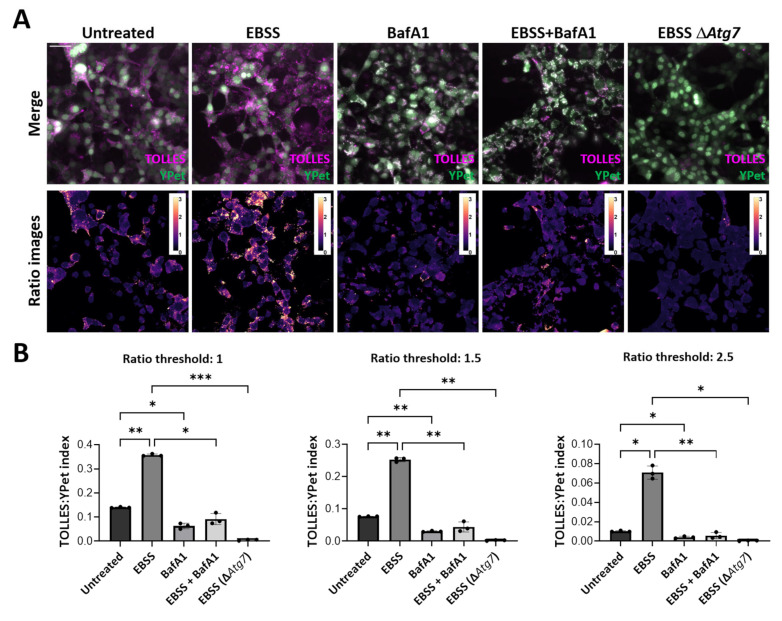
Quantification of SRAI ratio as a reliable measure of autophagic flux. (**A**) Representative immunofluorescence merge images of TOLLES (magenta) and YPet (green) from Figure 1C and their corresponding ratio images generated using the developed ImageJ macro. Bar = 50 µm. (**B**) Quantification of the TOLLES:YPet index from A using different set up thresholds employing ratios higher than 1; 1.5 and 2.5 (*n* = 3, 6 fields per replicate were analyzed with 60–100 cells per field, ±S.D., * = *p* < 0.05, ** = *p* < 0.01, *** = *p* < 0.001, 1-way ANOVA followed by Holm-Šídák comparisons).

**Figure 3 cells-12-01134-f003:**
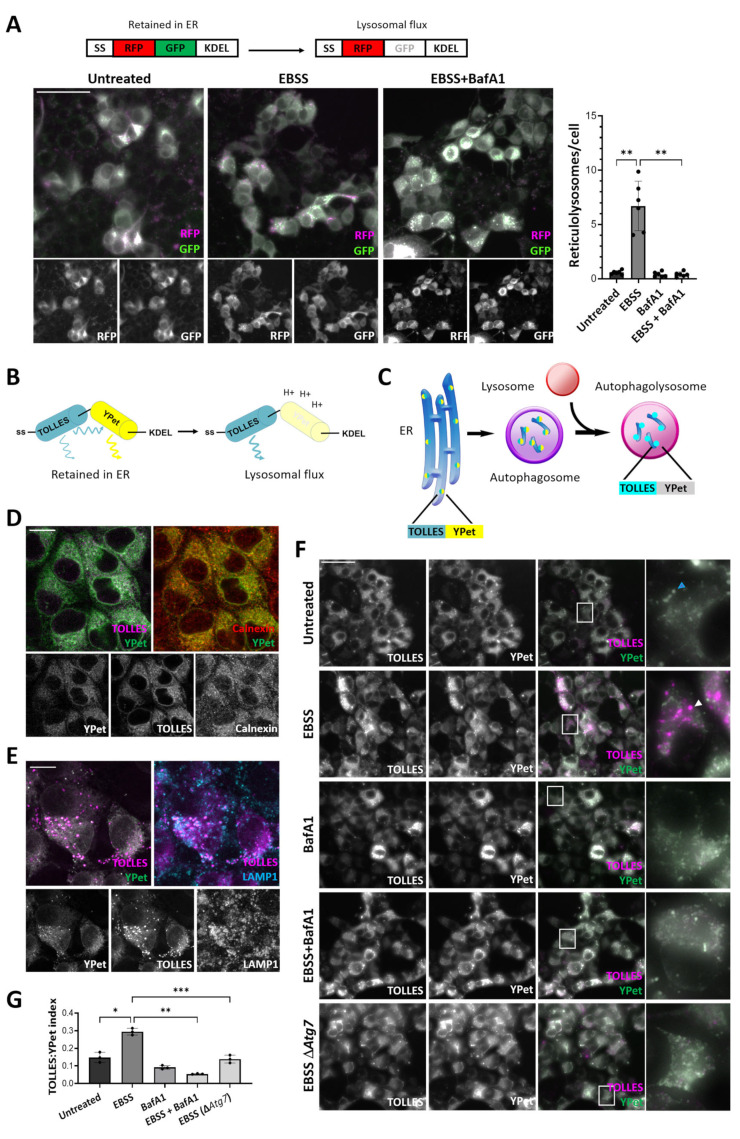
Development of ss-SRAI-KDEL for reliable measurement of general ER-phagy. (**A**) Schematic of the expected fluorescence profile of the ER-phagy reporter ss-RFP-GFP-KDEL within different compartments, where GFP is quenched in the lysosomes. RFP is represented in red and GFP in green or white if quenched. ss, signal sequence. Imaging-based autophagic flux assay in ss-mRFP-GFP-KDEL PDAC cells in full nutrients (untreated) or following 4 h starvation in the presence or absence of BafA1 (50 nM) and imaged using automated widefield microscopy. Cells were cultured in the presence of doxycycline (0.5 µg/mL, 24 h) to induce the reporter expression. Representative immunofluorescence images of RFP (magenta) and GFP (green). Bar = 50 µm. On the right, quantification of the number of reticulolysosomes (RFP-positive and GFP-negative) per cell scored using MitoQC-counter (*n* = 6, with 50–100 cells per replicate, ±S.D., ** = *p* < 0.01, 1-way ANOVA followed by Holm-Šídák comparisons). (**B**) Schematic of the expected fluorescence profile of SRAI (TOLLES-YPet) with an ER localization sequence (ss) in the N terminus and an ER retention KDEL sequence in the C terminus within different compartments (representing TOLLES as a FRET donor and YPet as a FRET acceptor). (**C**) ss-SRAI-KDEL reporter localizes in the ER or autophagosomes and TOLLES and YPet fluorescence is observed. However, after lysosomal fusion, the YPet is quenched and only TOLLES fluorescence is visible. (**D**,**E**) Immunofluorescence images of the ER marker Calnexin ((**D**), red) or the lysosomal marker LAMP1 ((**E**), blue) in full nutrient (**D**) or nutrient-starved (4 h, (**E**)) PDAC cells stably expressing ss-SRAI-KDEL (TOLLES, magenta; YPet, green) and imaged by confocal microscopy. Bar = 10 µm. (**F**) Imaging-based autophagic flux assay in ss-SRAI-KDEL PDAC cells (transfected with sgControl or sg*Atg7*) in full nutrients (untreated) or following 4 h starvation in the presence or absence of BafA1 (50 nM) and imaged using automated widefield microscopy. Representative immunofluorescence images of TOLLES (magenta) and YPet (green); white arrow shows an example of TOLLES-positive and YPet-negative puncta and blue arrow indicates TOLLES and YPet-positive puncta. Bar = 50 µm. (**G**) Quantification of the TOLLES:YPet index from (**F**) (*n* = 3, 6 fields per replicate were analyzed with 60–100 cells per field, ±S.D., * = *p* < 0.05, ** = *p* < 0.01, *** = *p* < 0.001, 1-way ANOVA followed by Holm-Šídák comparisons).

**Figure 4 cells-12-01134-f004:**
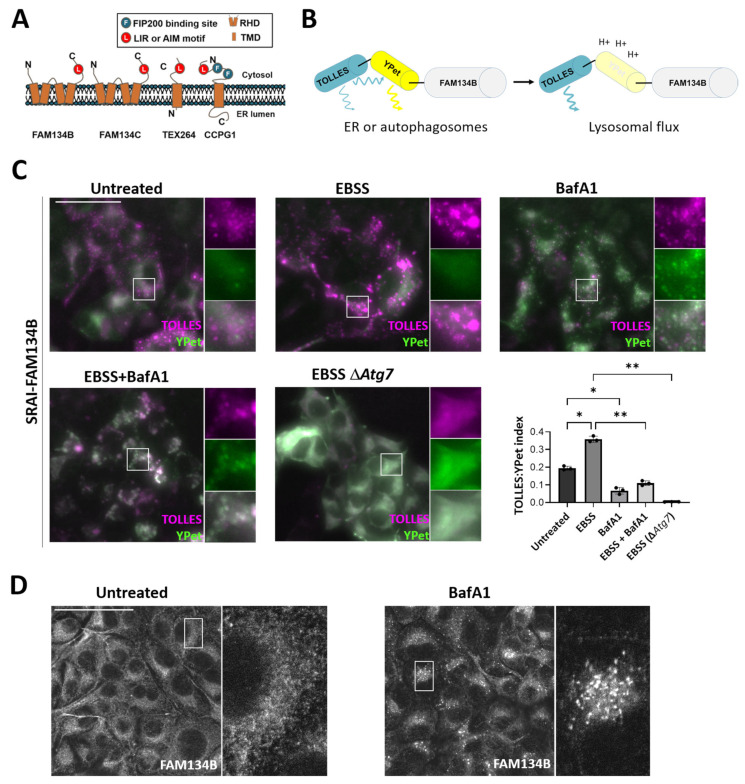
FAM134B flux can be monitored using the SRAI probe. (**A**) Schematic representing some of the characterized ER-phagy cargo receptors. LIR, LC3 interacting region; AIM, ATG8 interaction region; RHD, reticulon homology domain; TMD, transmembrane domain. (**B**) Schematic of the expected fluorescence profile of SRAI (TOLLES-YPet)-FAM134B within different compartments (representing TOLLES as a FRET donor and YPet as a FRET acceptor). TOLLES is represented in blue and YPet in yellow or faint yellow if quenched. (**C**) Imaging-based autophagic flux assay in PDAC cells stably expressing SRAI-FAM134B (transfected with sgControl or sg*Atg7*) in full nutrients (untreated) or following 4 h starvation in the presence or absence of BafA1 (50 nM) and imaged using automated widefield microscopy. Representative immunofluorescence images of TOLLES (magenta) and YPet (green). Bar = 50 µm. On the bottom right, quantification of the TOLLES:YPet index (*n* = 3, 6 fields per replicate were analyzed with 60–100 cells per field, ±S.D., * = *p* < 0.05, ** = *p* < 0.01, 1-way ANOVA followed by Holm-Šídák comparisons). (**D**) Immunofluorescence images of PDAC cells treated in the presence or absence of BafA1 (50 nM, 4 h). Cells were fixed and stained for FAM134B and imaged by confocal microscopy. Bar = 50 µm.

**Figure 5 cells-12-01134-f005:**
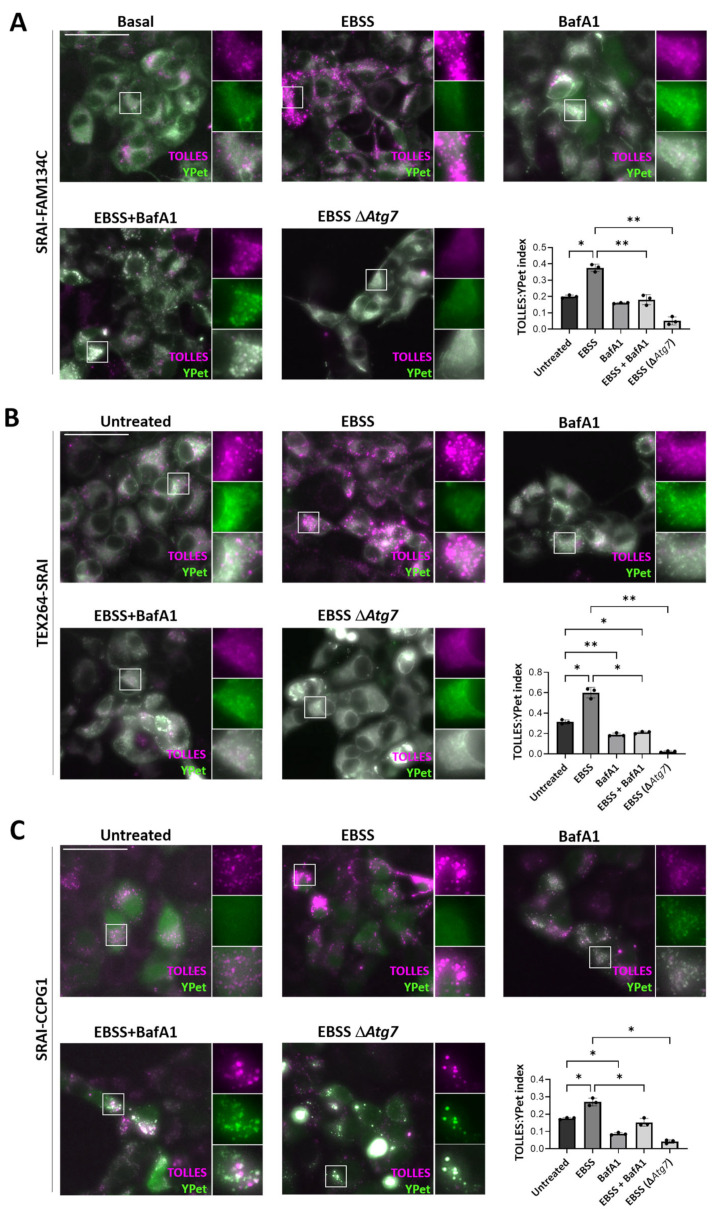
ER-phagy flux driven by specific ER-phagy receptors can be detected using the SRAI probe. (**A**–**C**) Imaging-based autophagic flux assay in PDAC cells stably expressing SRAI-FAM134C (**A**) or TEX264-SRAI (**B**) or SRAI-CCPG1 (**C**) (transfected with sgControl or *sgAtg7*) in full nutrients (untreated) or following 4 h starvation in the presence or absence of BafA1 (50 nM) and imaged using automated widefield microscopy. Representative immunofluorescence images of TOLLES (magenta) and YPet (green). Bar = 50 µm. On the bottom right, quantification of the TOLLES:YPet index (*n* = 3, 6 fields per replicate were analyzed with 60–100 cells per field, ±S.D., * = *p* < 0.05, ** = *p* < 0.01, 1-way ANOVA followed by Holm-Šídák comparisons).

## Data Availability

Data is available from the author. ImageJ macro and the JOB script are available from GitHub (accessed on 5 April 2023).

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
