# Peer review of "Signal-Retaining Autophagy Indicator as a Quantitative Imaging Method for ER-Phagy"

_cells, 2023, doi:10.3390/cells12081134_

Round 1
Reviewer 1 Report
In this manuscript, the authors validated the use of the SRAI and described the quantification methods. In addition, they developed new ER-phagy probes utilizing ER targeting sequences or ER-phagy receptors. The work described here is important and necessary in the field. However, further characterization of ER-phagy probes is strongly recommended before publication.
- The following sentence in the abstract is too long. The manuscript contains several sentences that are also long. Please review the sentences and make them concise.
In this study, we validate the use of the signal-retaining autophagy indicator (SRAI), a fixable fluorescent probe recently generated and described to detect mitophagy, as a versatile, sensitive and convenient probe for monitoring either general selective degradation of the endoplasmic reticulum (ER-phagy) or individual forms of ER-phagy involving specific cargo receptors (e.g. FAM134B, FAM134C, TEX264, and CCPG1).
- Fig.1C and 2A should be modified to make them more professional.
- Fig. 1D: The emission of TOLLES should be weak in the untreated group, however, I see a large number of magenta dots. It is possibly because the FRET may be weak in the configuration of TOLLES-short linker-YPet. The authors should provide direct evidence of FRET between TOLLES and YPet in these autophagy probes, in order to explain the mechanism of the probes with FRET.
- Fig. 1D: Starve ïƒ Starved
- Fig. 4: Overexpression of ER-phagy receptors in SRAI probes may interfere with the activity of endogenous receptors. Please provide data regarding this issue.
- Fig. 4: Since the subject of the paper is new probes for ER-phagy, further characterization of new probes would be required, for example monitoring of the progression of ER-phagy, colocalization with autophagy markers and so on. It would also be helpful if there were more comparisons between the different versions of the probes.
- Fig. 4: The size of the figure is too large. Maybe it can be divided into two figures.
Author Response
Reviewer 1
In this manuscript, the authors validated the use of the SRAI and described the quantification methods. In addition, they developed new ER-phagy probes utilizing ER targeting sequences or ER-phagy receptors. The work described here is important and necessary in the field. However, further characterization of ER-phagy probes is strongly recommended before publication.
We thank the reviewer for this careful and welcome analysis. Broadly, we agree with the changes that they have suggested but feel that we need to highlight that this study presents the use of SRAI probe, previously described for the study of mitophagy, in ER-phagy. We present it as a quantitative imaging method comparable to existing field-standard mRFP-GFP probes that are also generally overexpressed from an exogenous vector. In this vein, we have now provided new data with the existing ss-mRFP-GFP-KDEL reporter for general ER-phagy as a comparison to highlight this point and provide further characterization (Fig. 3A), as well as comparing the FAM134B overexpressed reporter behavior with endogenous FAM134B (Fig. 4D).
- The following sentence in the abstract is too long. The manuscript contains several sentences that are also long. Please review the sentences and make them concise.
In this study, we validate the use of the signal-retaining autophagy indicator (SRAI), a fixable fluorescent probe recently generated and described to detect mitophagy, as a versatile, sensitive and convenient probe for monitoring either general selective degradation of the endoplasmic reticulum (ER-phagy) or individual forms of ER-phagy involving specific cargo receptors (e.g. FAM134B, FAM134C, TEX264, and CCPG1).
This has now been addressed and several sentences in the manuscript have been modified accordingly. Please see tracked changes in the revised manuscript.
- Fig.1C and 2A should be modified to make them more professional.
We agree with this point. It is difficult to show the screenshots of the macro in operation in the limited space for print publication format. We have thus moved these figures to supplemental to indicate that they should not be viewed at print size but can be zoomed in upon and consulted where necessary (for those readers interested in seeing how the workflow we describe actually looks in practice).
- Fig. 1D: The emission of TOLLES should be weak in the untreated group, however, I see a large number of magenta dots. It is possibly because the FRET may be weak in the configuration of TOLLES-short linker-YPet. The authors should provide direct evidence of FRET between TOLLES and YPet in these autophagy probes, in order to explain the mechanism of the probes with FRET.
The paper describing the SRAI probe (Katayama et al., 2020) described the FRET mechanism and the properties of the fluorophores in detail when using this exact linker between the two fluorophores. The FRET interaction in cellulo does not completely prevent TOLLES fluorescence (when excited directly and emission imaged directly), rather quantitatively suppresses it.
Furthermore, depending on the level of basal (general or specific receptor-associated) ER-phagy activity in the untreated condition, the probe will still be targeted to autolysosomes to some extent (TOLLES only puncta) (although for all reporters this is greatly enhanced by starvation). We add new data to show this basal ER-phagy is apparently also seen with the field-standard ss-mRFP-GFP-KDEL probe (Fig. 3A).
- Fig. 1D: Starve à Starved
This was changed out as requested.
- Fig. 4: Overexpression of ER-phagy receptors in SRAI probes may interfere with the activity of endogenous receptors. Please provide data regarding this issue.
We have now included Fig. 4D to confirm the high basal turnover levels of FAM134B looking at endogenous levels in line with the results of the overexpressed SRAI-FAM134B reporter. Conversely, we still acknowledge that the overexpression in cells might affect the endogenous reporter. However, this limitation, which is now addressed in the text (Discussion-line 584), applies to overexpression of all mRFP-GFP based reporters as used routinely in the field. The focus of this study is to show the ease of use and a pathway for quantifiable data generation using a substitute (similar size) fluorescent probe with greater sensitivity, i.e. SRAI in place of mRFP-GFP, as reported by Katayama et al., 2020.
- Fig. 4: Since the subject of the paper is new probes for ER-phagy, further characterization of new probes would be required, for example monitoring of the progression of ER-phagy, colocalization with autophagy markers and so on. It would also be helpful if there were more comparisons between the different versions of the probes.
In this study, we focus on the specific use of SRAI for the monitor and quantification of ER-phagy and ER-phagy receptor flux to the lysosome end-point, as is standard for such probes. We have now included the field-standard ss-RFP-GFP-KDEL in Fig. 3A as a comparison with the ss-SRAI-KDEL.
We do agree that the autophagy proteins involved (or not) is of great interest. However, all such flux probes, existing (i.e. mRFP-GFP based, Halotag based etc.), or our newly described SRAI-based probes, really just measure the end-point of delivery to the lysosome, and while it is clear that this is autophagy dependent (i.e. as shown by Atg7 knockout), it is entirely possible other pathways could be involved under different stimuli. However, this is for future investigation. We have focused here on demonstrating a method for clear, high-content and quantitative measurement of flux (whether by classic macroautophagy or potentially other mechanisms of delivery of ER to the lysosome).
- Fig. 4: The size of the figure is too large. Maybe it can be divided into two figures.
This was carried out as requested.
Reviewer 2 Report
In the present manuscript, the authors presented a new tool to monitor macro-autophagy and the selective ER-phagy. This new system is based on the TOLLES/YPet ratio and data are elaborated using FIJI program. This tool has potentials and compensates for some deficits of the more traditional ones.
The experiments are properly performed and authors used solid protocols and controls. There are no major issues. However, few points should be taken into consideration.
1. A comparison between the new system and the classical RFP-GFP tandem reporter should be done to show that the results are in the same direction with some advantages. Authors may consider to perform some experiments using the RFP-GFP-KDEL and the TOLLES-YPet-KDEL to prove quality of the system
2. TOLLES-YPes looks quite a bulky tag. Authors should check that this tags is not forming ER luminal aggregates.
Author Response
In the present manuscript, the authors presented a new tool to monitor macro-autophagy and the selective ER-phagy. This new system is based on the TOLLES/YPet ratio and data are elaborated using FIJI program. This tool has potentials and compensates for some deficits of the more traditional ones.
We thank the reviewer for their positive comments and for their input to improving our manuscript we have now included the suggested experiments.
The experiments are properly performed and authors used solid protocols and controls. There are no major issues. However, few points should be taken into consideration.
- A comparison between the new system and the classical RFP-GFP tandem reporter should be done to show that the results are in the same direction with some advantages. Authors may consider to perform some experiments using the RFP-GFP-KDEL and the TOLLES-YPet-KDEL to prove quality of the system
We have now included an ss-RFP-GFP-KDEL in Fig. 3A as a comparison for the ss-SRAI-KDEL. The main advantages are in the ease of quantification of SRAI-KDEL with our pathway, the lack of background in the lysosome channel (TOLLES versus mRFP) for SRAI, and the ability to get statistically significant data from a small number of replicates (will be useful for higher throughput, screening assays).
- TOLLES-YPes looks quite a bulky tag. Authors should check that this tags is not forming ER luminal aggregates.
Using bulky tags are indeed a limitation for imaging assays and we now address this (Discussion Line 587). However, imaging of this construct in widefield and confocal microscopy after colocalization with calnexin (Figure 3D, E) does not show a disruption of the ER or accumulation of the probe in focal aggregates (nor was such a phenomenon reported in published data using this probe for mitophagy flux measurement, e.g Katayama et al 2020). In addition, mRFP-GFP-KDEL (with roughly a similar size tag) has been used in many studies. The main advantage we wish to highlight for this probe is the ability to easily perform assays with significant throughput, via semi-automated imaging. N.B. we acknowledge that Halotag systems may circumvent issues with tag size but throughput of these is limited (this is now discussed in new manuscript at Discussion Line 588).
Reviewer 3 Report
This article entitled “Signal-retaining autophagy indicator as a quantitative imaging method for ER-phagy” by Natalia Jimenez-Moreno develop a novel method to monitor ER-pahgy. They adopted and modified the signal-retaining autophagy indicator in Mitophagy assay study to bulk autophagy and ER-phagy.
The data are mostly clear.These methods successfully monitor the autophagy flux, which should be quite useful to the field.
Minor comment
1 Line 287. Establishing?
Author Response
This article entitled “Signal-retaining autophagy indicator as a quantitative imaging method for ER-phagy” by Natalia Jimenez-Moreno develop a novel method to monitor ER-pahgy. They adopted and modified the signal-retaining autophagy indicator in Mitophagy assay study to bulk autophagy and ER-phagy.
The data are mostly clear.These methods successfully monitor the autophagy flux, which should be quite useful to the field.
We thank the reviewer for their positive comments and for their input
Minor comment
1 Line 287. Establishing?
This has now been addressed.